# Disinfection by-Products and Ecotoxic Risk Associated with Hypochlorite Treatment of Tramadol

**DOI:** 10.3390/molecules24040693

**Published:** 2019-02-15

**Authors:** Valeria Romanucci, Antonietta Siciliano, Emilia Galdiero, Marco Guida, Giovanni Luongo, Renato Liguori, Giovanni Di Fabio, Lucio Previtera, Armando Zarrelli

**Affiliations:** 1Department of Chemical Sciences, University of Napoli Federico II, Via Cintia 4 (ed. 5), I-80126 Naples, Italy; valeria.romanucci@unina.it (V.R.); giovanni.luongo@unina.it (G.L.); difabio@unina.it (G.D.F.); previter@unina.it (L.P.); 2Department of Biology, University of Naples Federico II, via Cintia 4 (ed. 7), I-80126 Naples, Italy; antonietta.siciliano@unina.it (A.S.); egaldier@unina.it (E.G.); marco.guida@unina.it (M.G.); 3Department of Science and Technology, University of Naples Parthenope, I-80143 Naples, Italy; renato.liguori@uniparthenope.it

**Keywords:** tramadol, chlorination, disinfection by-products (DBPs), chlorine derivatives, disinfection treatments, acute and chronic toxicity tests

## Abstract

In recent years, many studies have highlighted the consistent finding of tramadol (TRA) in the effluents from wastewater treatment plants (WTPs) and also in some rivers and lakes in both Europe and North America, suggesting that TRA is removed by no more than 36% by specific disinfection treatments. The extensive use of this drug has led to environmental pollution of both water and soil, up to its detection in growing plants. In order to expand the knowledge about TRA toxicity as well as the nature of its disinfection by-products (DBPs), a simulation of the waste treatment chlorination step has been reported herein. In particular, we found seven new by-products, that together with TRA, have been assayed on different living organisms (*Aliivibrio fischeri, Raphidocelis subcapitata* and *Daphnia magna*), to test their acute and chronic toxicity. The results reported that TRA may be classified as a harmful compound to some aquatic organisms whereas its chlorinated product mixture showed no effects on any of the organisms tested. All data suggest however that TRA chlorination treatment produces a variety of DBPs which can be more harmful than TRA and a risk for the aquatic environment and human health.

## 1. Introduction

According to information available to the International Narcotics Control Board (INCB), the loose legislation in China, India and Sri Lanka allows mass production, not always legal, of tramadol (TRA). Generally, the drug arrives by ship to the Mediterranean where Italy and Greece are important points of disembarkation and distribution, but above all the main flow is towards Libya. The traffic then continues to Egypt and ends up in the main Middle Eastern countries, where in many countries it is not considered illegal. The powerful opioid, which in Europe can only be sold with a non-repeatable medical prescription, is called the “drug of the fighter”, costs a few euros per tablet and militias take it above all to endure the pain of wounds. It is estimated that world traffic drives a turnover of 1.39 billion dollars a year. The total amount of TRA used worldwide in the period from 1990 to 2009 was calculated to be 11,758 million of defined daily doses (1 DDD is equal to 300 mg) [1]. There is evidence of increasing traffic in TRA preparations to North and West Africa, as indicated by recent large seizures of such preparations in these regions. Egyptian authorities seized about 120 million tablets containing TRA in 2011 and about 320 million tablets in the first quarter of 2012. Saudi Arabia also reported increasing amounts of seizures of preparations containing TRA. In Gaza 2.5 million of TRA pills were seized in 2009, compared to 550,000 in 2008 [2]. In Benin, Ghana, Senegal and Togo (West Africa), large amounts of TRA preparations, totalling more than 132 tons of such preparations, were seized between February and October 2012. The preparations had been concealed in sea containers sent from India and were intercepted by the local law enforcement authorities [3]. In Wales, medical prescriptions are estimated at 2145 kg of product per year and as many as 25,000 kilos only in Germany [4,5].

TRA in 2013 was identified in a Cameron tree called *Sarcocephalus latifolia* [6]. Three years ago, some German and Cameroonian researchers demonstrated its anthropogenic nature through measurements of ^14^C content [7]. Evidently, the massive and illegal use as doping agent, both for humans and animals in order to withstand unsustainable workloads at temperatures above 40 °C, have led to consistent environmental pollution of water and soil. In some cases, the tests revealed the presence of TRA also in plants other than *S*. *latifolia* and in public wells, at alarming concentrations [8]. According to their physicochemical properties, TRA is expected to remain mainly in the water phase, considering its high water solubility and that volatilization is negligible because of the low Henry coefficients. Furthermore, its *n*-octanol/water partition coefficient indicates a tendency to remain in the water phase instead of accumulation in sewage sludge or in aquatic organisms. Could be interesting to consider the long-term side effects of TRA in humans, animals and the environment [9].

In recent studies, TRA has been detected in effluents from wastewater treatment plants (WWTPs) and in some rivers and lakes in both Europe and North America [10,11,12]. Several reports proved that 15–35% of the used product is excreted via the urine together with 15% of the desmethylated product, which is the major metabolite [10]. In general, the metabolic fate of TRA is quite complex with about 12 conjugated and 11 non-conjugated products [13,14,15]. The presence and fate of TRA and other 55 pharmaceuticals, endocrine disruptors, illicit drugs and personal care products (PPCPs) were investigated in the South Wales region of the UK [16], where it has been estimated an average consumption of 1654 mg/day per 1000 people, based on the data provided by the English National Health Service (NHS) of 2010 [12]. In this regard, it was reported that in the UK Cilfynydd WWTP influents, the maximum TRA concentrations was 89,026 ng/L and unexpectedly into the effluents was around 97,616 ng/L [16,17]. However, there is only a limited data available about the reduction of this compound and its metabolites in the WWTPs, estimating that no more than 36% of the TRA is removed by tricking filter and activated sludge treatment [18].

There are no precise data on the removal of TRA from wastewaters [19,20] and, in particular, by aqueous chlorination, only a paper by Cheng et al. [21] which reported the formation of a variety of chlorinated derivatives identified only by mass spectrometry. In general, during chlorination disinfection treatment, chorine reacts with natural organic matter and other organic compounds to form so-called disinfection by-products (DBPs) [22,23,24]. Several studies have shown that DBPs are usually more resistant to degradation and can be more toxic than their parent drugs [25,26,27,28]. In this paper, the degradation pathway of TRA was investigated by simulating the chlorination process normally used in a wastewater treatment plant to reduce similar emerging pollutants. The products obtained were identified and their acute and chronic toxicity assayed on different aquatic organisms (*Aliivibrio fischeri, Raphidocelis subcapitata* and *Daphnia magna*), using TRA for comparision.

## 2. Results and Discussion

### 2.1. Chlorination Data

Chlorination of TRA produced the disinfection by-products **BP1**–**BP7** (Scheme 1) that were isolated by chromatography and identified on the basis of their physical features. Compound **BP1** has the molecular formula C_16_H_24_ClNO_2_ according to the presence of peaks at *m*/*z* 297.81 [M]^+^ and 299.80 [M + 2]^+^ in the EI mass spectrum and sixteen carbon signals in the ^13^C-NMR spectrum. The mass spectrum showed also a fragment at *m*/*z* 262.37 due to the loss of a chlorine and a fragment at *m*/*z* 246.37 due to the loss of oxygen plus chlorine atoms bound to the C-1 benzyl carbon.

The NMR spectra resembled those of TRA. In particular, the ^1^H-NMR spectrum showed the four signals of four aromatic protons (δ 6.77, 7.05, 7.16 and 7.24) and, at high field, the characteristic signals of the methoxyl, H-2 and H-7 protons (δ 3.85, 1.84, 2.06/2.40, respectively) in addition to a peak at δ 2.13/2.15 integrable for six protons and related to the two methyls on the nitrogen atom. Assignments of protons to carbons were based on a HSQC experiment, while the HMBC spectra allowed the attribution of the signals of the two quaternary carbons to δ 152.00 and 159.98 to C-1′ and C-3′ carbons, respectively. The compound was identified as the hypochlorite corresponding to the starting product.

**BP2** has molecular formula C_14_H_21_NO_3_ according to the presence of the molecular ion at *m*/*z* 251.30 [M]^+^ in the EI mass spectrum and fourteen carbon signals in the ^13^C-NMR spectrum. In the ^1^H NMR the four aromatic ring protons seen at δ 6.81, 7.00, 7.05 and 7.29 correlated in the HSQC spectrum to the carbons at δ 111.97, 116.68, 110.88 and 129.26, respectively. The ^13^C-NMR spectrum, besides the signals of the protonated carbons, showed two quaternary carbons signals at δ 159.52 and 148.48 that HMBC experiments allowed to assign the first one to C-3′ and the latter to C-1′. The DEPT spectrum showed the presence of two methines at δ 2.15 and 4.45, identified as H-2 and H-7, respectively, whose protons were correlated with each other in the COSY spectrum and both correlated to C-1 carbon in the HMBC spectrum, at δ 77.07. In the last spectrum the methine H-2 was correlated to the aromatic carbon C-1′, as well as to the C-3, C-6 and C-7 carbons, at δ 26.23, 40.38 and 67.85 respectively, while the methine H-7 was correlated to C-2 and C-3, at δ 32.70 and 26.23, respectively.

**BP3** had a pseudo-molecular ion at *m*/*z* 234.35 [M]^+^ in the EI mass spectrum and fourteen carbon signals in the ^13^C-NMR spectrum, which correspond to the molecular formula C_14_H_18_O_3_. The ^1^H- NMR indicated the absence of the three carbons bounded to the nitrogen and the presence of two singlet signals at δ 9.61 e 6.65. In the HSQC spectrum these last two signals were correlated to the carbons at δ 192.90 e 150.02, respectively. In the HMBC spectrum the proton a δ 9.61 was correlated to the signals at δ 150.02, 143.50 e 21.44, identified as C-2, C-3 and C-4 carbons, respectively, while the proton a δ 6.65 was correlated to the carbons C-3 and C-4 and to the signals at δ 192.90 and 77.10, identified as carbons C-7 and C-1, respectively.

**BP4** had a molecular ion at *m*/*z* 261.34 [M]^+^ in the EI mass spectrum and sixteen carbon signals in the ^13^C-NMR spectrum, which correspond to the molecular formula C_16_H_23_NO_2_. The ^1^H-NMR indicated the absence of one of the two methyls bound to the nitrogen and the presence of a signal at δ 3.62 and 4.20, correlated in a HSQC spectrum at the carbon δ 82.40 (C-10). The DEPT spectrum confirmed the methylene nature of this signal and the HMBC spectrum indicated the correlation of the protons with the signals at δ 77.10, 56.12 and 40.36, identified as the C-1, C-7 and C-9 carbons, respectively, and the correlation of the corresponding carbon with the H-9 methyl protons at δ 2.01 and with the H-7 protons at δ 2.42 and 2.54.

**BP5** had a molecular ion at *m*/*z* 342.79 [M]^+^ and 344.80 [M + 2]^+^ in the EI mass spectrum and seventeen carbon signals in the ^13^C-NMR spectrum, which correspond to the molecular formula C_17_H_23_ClO_5_. The ^1^H-NMR showed the presence of four protons of aromatic ring and a methoxyl groups. The ^13^C-NMR spectrum showed the presence of the signals of the aromatic ring and of its methoxyl group as well as other ten signals, of which two carbonyls at δ 199.69 and 208.13. The remaining height signals in a DEPT experiment were identified as one methyl, five methylenes, one methine and one quaternary carbon, whose corresponding protons were identified by an HSQC spectrum. The carbon at δ 199.69, identified as C-1, was correlated in an HMBC spectrum to the two aromatic protons H-2′ and H-6′, at δ 7.51 and 7.56, respectively, and to the signals at δ 3.04 and 1.83, identified as protons H-2 and H-3, respectively, that were coupled together in a COSY spectrum. The COSY spectrum also showed the correlation between the signals at δ 1.80 and 2.35, of which the first one was identified as relative to the protons H-4, correlated in the same COSY spectrum to the protons H-3, and the second one, identified as relative to protons H-5. The protons H-5 were correlated in a HMBC experiment to signals at δ 96.27 and 74.07, identified as carbons C-6 and C-7, respectively. In the same HMBC spectrum the quaternary carbon at δ 208.13 was correlated with the signal of the methyl protons at δ 2.28, with those of the methylene at δ 2.95 and 3.18 and with the methine at δ 4.39, identified as protons H-10, H-8 and H-7, respectively.

**BP6** had a molecular ion at *m*/*z* 277.38 [M]^+^ in the EI mass spectrum and sixteen carbon signals in the ^13^C-NMR spectrum, which could correspond to the molecular formula C_16_H_23_NO_3_. The ^1^H- NMR indicated the presence of a signal of an aldehyde function at δ 7.79 and the absence of one of the two methyls bound to the nitrogen. The HSQC spectrum showed the correlation of the aforementioned proton at low fields with carbon at δ 163.13 and the HMBC spectra showed its correlations with the signals at δ 50.68 and 29.76, identified as the carbons C-7 and C-9, respectively, confirming that the isolated product was the oxidized TRA to one of the two methyls.

**BP7** had a molecular ion at *m*/*z* 276.31 [M]^+^ in the EI mass spectrum, beyond that 246.30 and 219.29 due to the loss of NO and NO and HCN; moreover the ^13^C-NMR spectrum showed fifteen carbon signals, which could correspond to the molecular formula C_15_H_20_N_2_O_3_. The ^1^H-NMR indicated the absence of the two methyls bound to the nitrogen, confirmed the presence of the signals of aromatic ring and of methoxyl group and the presence of two pairs of signals at δ 2.66 and 2.70 and at δ 2.73 and 2.75, correlated in an HSQC spectrum to the signals at 67.40 and 54.00, respectively. In correspondence of these last two signals the DEPT spectrum confirmed the presence of two methylenes, as well as two methines at δ 44.53 and 25.81, bound in the HSQC spectra to the protons at δ 2.15 and 1.95, respectively. The methylene protons at δ 2.66/2.70 were correlated in an HMBC spectrum with the signals a δ 25.81, 44.53 and 26.11, identified as the carbons C-3, C-2 and C-4, while the methylene protons at δ 2.73/2.75 were correlated with the carbons C-2 and C-3 and with the signal a δ 77.50, identified as the carbon C-1. A possible mechanism for obtaining the products **BP1**–**BP7** is shown in Scheme 1.

### 2.2. Ecotoxicity Data

Ecotoxicity is essential to determine the potential hazard and risk of emerging contaminants. Based on literature reviews, TRA-associated ecotoxicity is scarcely reported [29] and its transformation by-products after chlorination have not been reported elsewhere. Published findings [29,30] indicated that TRA induces only negligible effects with half maximal effective concentration (EC_50_) values up to 100 mg/L. Bergheim et al. [30] found that a concentration of 374 mg/L TRA induced a 50% growth inhibition in *Pseudomonas putida* bacterial strain. Another study showed slight TRA toxicity in *Daphnia magna* [29] with an EC_50_ value of 170 mg/L. Effect concentration (EC_50_) of TRA, its chlorination mixture and disinfection by-products on *A. fischeri*, *R. subcapitata* and *D. magna* organisms is presented in Table 1. Considering the risk classification system on EC_50_ values [31], toxicity data (*D. magna* (24 h), *R. subcapitata* and *A. fischeri*) were combined and ranked in four main groups of samples as: (i) very toxic (EC_50_ < 1 mg/L); (ii) toxic (EC_50_ 1–10 mg/L); harmful (EC_50_ 10–100 mg/L); and (iii) not harmful (EC_50_ >100 mg/L). Results of parental compound showed that the relative sensitivity order was *R. subcapitata > D. magna > A. fischeri*. In particular, no toxic effect was observed for TRA with *A. fischeri*. In fact, luminescence was measurably enhanced during contact with TRA, probably because it was consumed as a nutrient by the bacteria. The highest TRA concentration used was 100 mg/L, thus it may be inferred that EC_50_ value exceeds 100 ppm. TRA-associated EC_50_ was estimated as 88.5 mg/L and 87.1 mg/L for *D. magna* and *R. subcapitata* respectively. According to the EU Directive 93/67/EEC [31], the TRA may be classified as harmful to aquatic organisms, as according to the European regulation EC_50_ values less than 100 mg/L gives a classification of “Harmful” to aquatic organisms [32]. This means that TRA pseudopersistence will be a concern, and in some cases, it might have toxicity.

A comparison of our EC_50_ value for TRA in *D. magna* with published data [29] has been done, but some discrepancies were observed, the mismatch being probably due to differences between the test protocols used [33,34]. The same bioassays were carried out to assess the ecotoxicological effects of TRA chlorinated mixture and the obtained data demonstrated that no effect was induced in bacteria, while micro-crustaceans and algae were more sensitive to the mixture than the parent compound with EC_50_ values of about 50 mg/L. A comparison among the adverse effects of TRA mixture, denotes that TRA degradation does not follow a toxicity decrease with a possible synergistic interaction between the byproducts. In addition, we performed ecotoxicity tests for the **BP1**–**BP7** isolated products and we found a multiple effect of chlorination on the ecotoxicity profile in some of these byproducts; we also noticed a higher toxicity in *A. fischeri* compared to TRA as well as the mixture.

Products **BP6** and **BP2** were less toxic than TRA, with EC_50_ values ranged to 250 mg/L and 450 mg/L and could be classified as a not harmful compounds. **BP4**, **BP5**, **BP1, BP3** and **BP7** were found more toxic compared to TRA, suggesting that the adverse effects of oxidized species depends on the extent of oxidation [35]. According to the EU risk classification **BP4, BP5, BP1, BP3** and **BP7** are classified as harmful compounds. The relative toxicity order was **BP4** > **BP5** > **BP1** > **BP3** > **BP7**. In particular, the EC_50_ data were ranging 81–117 mg/L for **BP7**, 26–66 mg/L for **BP1** and 28–34 mg/L for **BP4**. In these cases, the degree of oxidation was also found to increase the toxicity of by-products suggesting that TRA oxidation has the potential to produce by-products, which can be more harmful than TRA.

## 3. Materials and Methods

### 3.1. Chemicals

TRA (**1**, 99.3%) was purchased from Sigma Aldrich (Milan, Italy). All the other chemicals and solvents were purchased from Fluka (Saint-Quentin Fallavier, France), with HPLC grade and were used as received.

### 3.2. Chlorination Reaction

#### 3.2.1. Apparatus

Column chromatography (CC) was carried out on Kieselgel 60 (230–400 mesh, Merck, Darmstadt, Germany). HPLC was performed on a Shimadzu LC-8A system using a Shimadzu SPD-10A VP UV-VIS detector (Shimadzu, Milan, Italy). Semipreparative HPLC was performed using a RP Gemini C18-110A preparative column (10 μm particle size, 250 mm × 21.2 mm i.d., Phenomenex, Bologna, Italy) column with a flow rate of 5.0 mL min^−1^. ^1^H- and ^13^C-NMR spectra were recorded on an INOVA-500 NMR instrument (Varian, Milan, Italy, ^1^H at 499.6 MHz and ^13^C at 125.62 MHz), referenced in ppm to residual solvent signals (CDCl_3_, at δ*_H_* 7.27 and δ*_C_* 77.0) at 25 °C. Proton-detected heteronuclear correlations were measured using a gradient heteronuclear single-quantum coherence (HSQC), optimized for ^1^*J*_HC_ = 155 Hz, a gradient heteronuclear multiple bond coherence (HMBC), optimized for ^n^*J*_HC_ = 8 Hz. Electronic Impact Mass Spectra (EI-MS) were obtained with a QP-5050A EI 70 eV spectrometer (Shimadzu).

#### 3.2.2. Chlorination Procedure and Product Isolation

TRA (1 g, 6.17 mmol) dissolved in milliQ water (1 L) was treated for 30 min with 10% hypochlorite (molar ratio TRA: hypochlorite 1:6; concn, spectroscopically determined λ_max_ 292 nm, ε 350 dm^3^/mol cm) at room temperature [36]. The pH of the solution, measured by a pH-meter at five minute intervals, rose from the initial pH 8.5 to 9.8, after 5 min, and it remained at this value during the reaction time. After 30 min, the solution, quenched by sodium sulfite excess, was concentrated by lyophilization and extracted with ethyl acetate. The ethyl acetate fraction (EA) was chromatographed on silica gel CC using a gradient of petroleum ether:acetone (100:0 to 0:100), to give 22 fractions. The fraction EA3, eluted with petroleum ether:acetone (85:15), was rechromatographed on silica gel CC eluting with petroleum ether:acetone (90:10) to give nine subfractions. The subfractions EA1.2 and EA1.3 contained the by-products **BP1** and **BP2** (60 and 20 mg, respectively). The fraction EA20, eluted with petroleum ether:acetone (50:50), was analyzed by HPLC using a reversed phase column and eluting with methanol:acetonitrile:water (4:1:5), to give four subfractions. The subfraction EA20.1, EA20.2 and EA20.4 contained the byproduct **BP3**–**BP5** (6, 40 and 13 mg, respectively). The fraction obtained by extraction with water (W) was filtered on C18 silica gel CC with methanol and water (Scheme 2). The methanol fraction (WM) was chromatographed on silica gel CC using a gradient of methylene chloride:acetone (95:5 to 80:20), to give 9 fractions. The fraction WM1, eluted with methylene chloride:acetone (95:5), contained the byproduct **BP6** (17 mg) that was purified on TLC eluted with petroleum ether:acetone (85:15). The fraction WM5, eluted with methylene chloride:acetone (85:15), contained the byproduct **BP7**. It was rechromatographed on silica gel CC eluting with a hexane:acetone gradient (100:0 to 65:35) to give 7 subfractions. The subfraction WM5.5 contained the byproduct **BP7** (31 mg).

### 3.3. Spectral Data

*2-[(Dimethylamino)methyl]-1-(3-methoxyphenyl)cyclohexyl hypochlorite* (**BP1**). White powder. ^1^H-NMR (499.6 MHz, CDCl_3_): δ 1.35 (m, 1H, H-3), 1.55 (m, 1H, H-4), 1.65 (m, 2H, H-4, H-6), 1.75 (m, 1H, H-5), 1.80 (m, 2H, H-3, H-5), 1.84 (m, 1H, H-2), 2.06 (d, *J* = 13.70, 2.40 Hz, 1H, H-7), 2.08 (m, 1H, H-6), 2.13 (s, 3H, H-9), 2.15 (s, 3H, H-10), 2.40 (d, J = 13.70, 4.50 Hz, 1H, H-7), 3.85 (s, 3H, OCH_3_), 6.77 (dd, *J* = 8.2, 2.4 Hz, 1H, H-4′), 7.05 (dd, *J* = 8.0, 2.2 Hz, 1H, H-6′), 7.16 (br s, 1H, H-2′), 7.24 (dd, *J* = 8.2, 8.0 Hz, 1H, H-5′). ^13^C-NMR (125.62 MHz, CDCl_3_): δ 22.24 (C-5), 26.82 (C-3), 27.92 (C-4), 44.73 (C-2), 41.31 (C-6), 55.00 (OCH_3_), 61.55 (C-7), 77.00 (C-1), 47.72 (C-9, C-10), 111.25 (C-4′), 110.98 (C-2′), 117.36 (C-6′), 128.94 (C-5′), 152.00 (C-1′), 159.98 (C-3′). ESI-MS (positive ions): *m*/*z* calculated for C_16_H_24_ClNO_2_
*m*/*z* 297.82 [M]^+^ and 299.82 [M + 2]^+^; found 297.81 [M]^+^ and 299.80 [M + 2]^+^, 262.37 [M − Cl]^+^, 246.37 [M − OCl]^+^.

*2-[Amino(hydroxyl)methyl]-1-(3-methoxyphenyl)cyclohexanol* (**BP2**). White powder. ^1^H-NMR (499.6 MHz, CDCl_3_): δ 1.18 (m, 1H, H-5), 1.26 (m, 1H, H-5), 1.43 (m, 1H, H-3), 1.54 (m, 1H, H-4), 1.65 (m, 1H, H-6), 1.75 (m, 1H, H-4), 1.85 (m, 1H, H-3), 2.01 (m, 1H, H-6), 2.15 (m, 1H, H-2), 3.83 (s, 3H, OCH_3_), 4.45 (m, 1H, H-7), 6.81 (dd, *J* = 8.2, 2.4 Hz, 1H, H-4′), 7.00 (dd, *J* = 8.0, 2.2 Hz, 1H, H-6′), 7.05 (br s, 1H, H-2′), 7.29 (dd, *J* = 8.2, 8.0 Hz, 1H, H-5′). ^13^C-NMR (125.62 MHz, CDCl_3_): δ 20.91 (C-4), 26.23 (C-3), 26.95 (C-5), 32.70 (C-2), 40.38 (C-6), 55.21 (OCH_3_), 67.85 (C-7), 77.07 (C-1), 110.88 (C-2′), 111.97 (C-4′), 116.68 (C-6′), 129.26 (C-5′), 148.48 (C-1′), 159.52 (C-3′). ESI-MS (positive ions): *m*/*z* calculated for C_14_H_21_NO_3_ 251.32 [M]^+^; found 251.30 [M]^+^.

*2-Hydroxy-2-(3-methoxyphenyl)cyclohexanecarbaldehyde* (**BP3**). White powder. ^1^H-NMR (499.6 MHz, CDCl_3_): δ 1.63 (m, 1H, H-5), 1.72 (m, 1H, H-5), 2.03 (m, 3H, H-4, H-4, H-6), 2.31 (m, 1H, H-6), 3.85 (s, 3H, OCH_3_), 6.65 (s, 1H, H-2), 6.86 (dd, *J* = 7.8, 2.6 Hz, 1H, H-4′), 6.98 (dd, *J* = 7.8, 1.8 Hz, 1H, H-6′), 7.02 (br t, *J* = 1.9 Hz, 1H, H-2′), 7.31 (t, *J* = 8.0 Hz, 1H, H-5′), 9.61 (s, 1H, H-7). ^13^C-NMR (125.62 MHz, CDCl_3_): δ 18.40 (C-5), 21.44 (C-4), 39.28 (C-6), 55.32 (OCH_3_), 77.10 (C-1), 111.34 (C-2′), 112.74 (C-4′), 117.61 (C-6′), 129.58 (C-5′), 143.50 (C-3), 150.71 (C-1′), 150.02 (C-2), 159.95 (C-3′), 192.90 (C-7). ESI-MS (positive ions): *m*/*z* calculated for C_14_H_18_O_3_ 234.29 [M]^+^; found 234.35 [M]^+^, 205.37 [M − CHO]^+^.

*(4aS,8aS)-8a-(3-Methoxyphenyl)-3-methyloctahydro-2H-benzo[e][1,3]oxazine* (**BP4**). White powder. ^1^H-NMR (499.6 MHz, CDCl_3_): δ 1.30 (m, 1H, H-3), 1.50 (m, 1H, H-5), 1.65 (m, 1H, H-4), 1.70 (m, 1H, H-5), 1.80 (m, 1H, H-3), 1.90 (m, 1H, H-6), 1.94 (m, 1H, H-6), 2.01 (s, 3H, H-9), 2.31 (m, 1H, H-2), 2.38 (m, 1H, H-4), 2.42 (d, *J* = 13.0 Hz, 1H, H-7), 2.54 (d, *J* = 13.0 Hz, 1H, H-7), 3.62 (d, *J* = 8.6 Hz, 1H, H-10), 3.81 (s, 3H, OCH_3_), 4.20 (d, J = 8.6 Hz, 1H, H-10), 6.78 (dd, *J* = 8.2, 2.4 Hz, 1H, H-4′), 6.94 (dd, *J* = 8.0, 2.2 Hz, 1H, H-6′), 6.96 (br s, 1H, H-2′), 7.27 (dd, *J* = 8.2, 8.0 Hz, 1H, H-5′). ^13^C-NMR (125.62 MHz, CDCl_3_): δ 22.07 (C-5), 25.54 (C-3), 27.59 (C-4), 35.13 (C-2), 40.36 (C-9), 42.70 (C-6), 55.20 (OCH_3_), 56.12 (C-7), 77.10 (C-1), 82.40 (C-10), 111.66 (C-4′), 112.18 (C-2′), 118.25 (C-6′), 129.54 (C-5′), 147.30 (C-1′), 160.00 (C-3′). ESI-MS (positive ions): *m*/*z* calculated for C_16_H_23_NO_2_ 261.36 [M]^+^; found 261.34 [M]^+^.

*6-Chloro-6,7-dihydroxy-1-(3-methoxyphenyl)decane-1,9-dione* (**BP5**). White powder. ^1^H-NMR (499.6 MHz, CDCl_3_): δ 1.80 (m, 2H, H-4), 1.83 (m, 2H, H-3), 2.28 (s, 3H, H-10), 2.35 (m, 2H, H-5), 2.95 (dd, *J* = 18.0, 9.0 Hz, 1H, H-8), 3.04 (t, *J* = 6.5 Hz, 2H, H-2), 3.18 (dd, *J* = 18.0, 2.0 Hz, 1H, H-8), 4.39 (d, *J* = 8.9 Hz, 1H, H-7), 3.88 (s, 3H, OCH_3_), 7.13 (dd, *J* = 8.0, 2.4 Hz, 1H, H-4′), 7.40 (t, *J* = 8.0 Hz, 1H, H-5′), 7.51 (br s, 1H, H-2′), 7.56 (dd, *J* = 8.0, 2.2 Hz, 1H, H-6′). ^13^C-NMR (125.62 MHz, CDCl_3_): δ 23.62 (C-4), 24.47 (C-3), 30.87 (C-10), 38.31 (C-2), 43.60 (C-5), 45.30 (C-8), 55.50 (OCH_3_), 74.07 (C-7), 96.27 (C-6), 112.40 (C-2′), 119.50 (C-4′), 120.71 (C-6′), 129.62 (C-5′), 138.28 (C-1′), 159.87 (C-3′), 199.69 (C-1), 208.13 (C-10). ESI-MS (positive ions): *m*/*z* calculated for C_17_H_23_ClO_5_ 342.81 [M]^+^ and 344.81 [M + 2]^+^; found 342.79 [M]^+^ and 344.80 [M + 2]^+^.

*N-[(2-Hydroxy-2-(3-methoxyphenyl)cyclohexyl)methyl]-N-methylformamide* (**BP6**). White powder. ^1^H-NMR (499.6 MHz, CDCl_3_): δ 1.31 (m, 1H, H-3), 1.41 (m, 1H, H-5), 1.58 (m, 1H, H-3), 1.60 (m, 1H, H-6), 1.70 (m, 2H, H-4, H-5), 1.75 (m, 1H, H-6), 1.90 (m, 1H, H-4), 2.26 (m, 1H, H-2), 2.59 (s, 3H, H-9), 2.71 (d, *J* = 14.80, 3.00 Hz, 1H, H-7), 3.08 (d, *J* = 14.80, 10.40 Hz, 1H, H-7), 3.79 (s, 3H, OCH_3_), 6.73 (dd, *J* = 8.2, 2.4 Hz, 1H, H-4′), 7.01 (dd, *J* = 8.0, 2.2 Hz, 1H, H-6′), 7.18 (br s, 1H, H-2′), 7.26 (dd, *J* = 8.2, 8.0 Hz, 1H, H-5′), 7.79 (s, 1H, H-10). ^13^C-NMR (125.62 MHz, CDCl_3_): δ 21.80 (C-5), 25.41 (C-3), 25.63 (C-4), 29.76 (C-9), 41.54 (C-2), 43.30 (C-6), 50.68 (C-7), 55.23 (OCH_3_), 74.18 (C-1), 111.47 (C-2′), 111.72 (C-4′), 116.85 (C-6′), 129.45 (C-5′), 150.08 (C-1′), 159.74 (C-3′), 163.13 (C-10). ESI-MS (positive ions): *m*/*z* calculated for C_16_H_23_NO_3_ 277.36 [M]^+^; found 277.38 [M]^+^.

*4-(3-Methoxyphenyl)-2-nitrosooctahydro-1H-isoindol-4-ol* (**BP7**). White powder. ^1^H-NMR (499.6 MHz, CDCl_3_): δ 1.62 (m, 1H, H-5), 1.65 (m, 2H, H-4, H-6), 1.70 (m, 1H, H-5), 1.80 (m, 1H, H-6), 1.90 (m, 1H, H-4), 1.95 (m, 1H, H-3), 2.15 (m, 1H, H-2), 2.66 (m, 1H, H-9), 2.70 (m, 1H, H-9), 2.73 (d, *J* = 14.0 Hz, 1H, H-7), 2.75 (d, *J* = 14.0 Hz, 1H, H-7), 3.82 (s, 3H, OCH_3_), 6.77 (dd, *J* = 8.2, 2.4 Hz, 1H, H-4′), 7.03 (dd, *J* = 8.0, 2.2 Hz, 1H, H-6′), 7.06 (br s, 1H, H-2′), 7.28 (dd, *J* = 8.2, 8.0 Hz, 1H, H-5′). ^13^C-NMR (125.62 MHz, CDCl_3_): δ 21.96 (C-5), 25.81 (C-3), 26.11 (C-4), 44.53 (C-2), 41.62 (C-6), 55.16 (OCH_3_), 54.00 (C-7), 77.50 (C-1), 67.40 (C-9), 111.59 (C-4′), 110.71 (C-2′), 117.05 (C-6′), 129.13 (C-5′), 149.66 (C-1′), 160.22 (C-3′). ESI-MS (positive ions): *m/z* calculated for C_15_H_20_N_2_O_3_ 276.33 [M]^+^; found 276.31 [M]^+^, 246.30 [M − NO]^+^, 219.29 [M – NO − HCN]^+^.

### 3.4. Ecotoxicity Assays

TRA and its products were dissolved in 0.01% dimethyl sulfoxide (DMSO) and further diluted in synthetic freshwater to make stock solutions [34]. The acute bioluminescence inhibition assay was carried out using *Aliivibrio fischeri* (NRRL-B-11177), according to ISO 2007 [37]. The luminescence was measured with a Microtox^®^ analyzer (Model 500, AZUR Environmental, Carlsbad, CA, USA) after 30 min at 15 °C. Tests were carried out in triplicate. Data were analyzed with Microtox Omni software (version 1.18AZUR Environmental, Carlsbad, CA, USA) and the result expressed as percentage of bioluminescence inhibition (%). The chronic growth inhibition test with *Raphidocelis subcapitata* was carried out according to ISO 2012 [38]. The cultures were kept in Erlenmeyer flasks. The initial inoculum contained 104 cells/mL. The specific growth inhibition rate was calculated considering 3 replicates exposed at 24 ± 1 °C for 72 h under continuous illumination (120–60 μein/m^2^s). Effect data were expressed as percentage of growth inhibition [39].

Acute toxicity tests with *Daphnia magna* were carried out according to ISO 2007 [37]. Newborn daphnids (<24 h old) were exposed in four replicates for 24 and 48 h at 20 ± 1 °C in the dark. Toxicity was expressed as the percentage of dead organisms [40]. Toxicity was expressed as the median effective concentration (EC_50_) along with 95% confidence limit values considering parametric or non-parametric methods. Two-way analysis of variance was carried out followed by Tukey’s post hoc analysis after testing normality and homoscedasticity of data distributions. Statistical analysis and graphs were done using Microsoft^®^ Excel 2013/XLSTAT©-Pro (Version 7.2, 2003, Addin soft, Inc., Brooklyn, NY, USA).

## 4. Conclusions

This paper investigated the fate of TRA following disinfection treatment by chlorination and the ecotoxicity of this drug and its disinfection by-products. The reaction was carried out simulating the conditions of a typical wastewater plant, using an excess of sodium hypochlorite. TRA undergoes more than 70% of degradation but the treatment did not give rise the mineralization but rather induces the transformation of TRA itself into at least seven by-products, two of them chlorinated. Consistent with observations of previous studies based on toxic risk associated with hypochlorite treatment [22,23,24,40], the transformation by-products of TRA were confirmed to present higher toxicity than TRA. In particular, these data suggest that TRA chlorination has the potential to produce by-products which are more harmful than TRA. Considering the large consumption of this drug in both Europe and America, and the consistent amounts of it found in many effluents and also in lakes and rivers, it is imperative to consider the chlorination of TRA hazardous for its environmental impact and for human health. For a complete degradation of this pollutant it could be useful a longer chlorination process were applied or it were performed twice [22].

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
