# Peer review of "Disinfection by-Products and Ecotoxic Risk Associated with Hypochlorite Treatment of Tramadol"

_molecules, 2019, doi:10.3390/molecules24040693_

Round 1

Reviewer 1 Report

This paper provides an aspect related to the possible fate of painkiller medicine little known to the public and scientists as well. The manuscript is well written with clarity and the experiments and results are well executed and clear.

However, a few minor points should be addressed before publication.

1.    The audience may misunderstand that there is a waste treatment specifically designed fort tramadol, which is not. The introduction needs to clarify this point. In other words, WWT is just performed as guided by law and regulation and not for Trammadol.

2.     In relation to the point above, how can you prevent the DBPs from forming at WWT plants given the general purpose of chlorination, which is not targeted for Trammdol specifically?

3.    There is NO section for discussion. Is this part of the general policy of the journal?

4.    English needs to be improved and I suggest inviting a native speaker to read through as I (a non-native English user) always do for my publication. For instance, “have been highlighted” in abstract --> have highlighted  

Author Response

Dear Reviewer,

as you can notice, the comments and/or suggestions have been taken in due consideration. Our reply and/or our comments to the suggestions are in italics. In the manuscript the revisions are highlighted in yellow. More in detail:

Comments and Suggestions for Authors by Reviewer #1

This paper provides an aspect related to the possible fate of painkiller medicine little known to the public and scientists as well. The manuscript is well written with clarity and the experiments and results are well executed and clear.

However, a few minor points should be addressed before publication.

-          The audience may misunderstand that there is a waste treatment specifically designed fort Tramadol, which is not. The introduction needs to clarify this point. In other words, WWT is just performed as guided by law and regulation and not for Tramadol.

-          In relation to the point above, how can you prevent the DBPs from forming at WWT plants given the general purpose of chlorination, which is not targeted for Tramadol specifically?

-          There is NO section for discussion. Is this part of the general policy of the journal?

-          English needs to be improved and I suggest inviting a native speaker to read through as I (a non-native English user) always do for my publication. For instance, “have been highlighted” in abstract --> have highlighted

Replay: Done! In particular we have revised the introduction making the suggested clarifications. We have brought together the results and the discussion in a single paragraph and corrected some words (the corrections are reported as word-office revisions).

Reviewer 2 Report

- The authors should check the references listed in the text

- The results of ecotoxicology tests could be presented graphically (suggestion)

- some parts of the text shold be corrected (see attachment)

Author Response

Dear Reviewer,

as you can notice, the comments and/or suggestions have been taken in due consideration. Our reply and/or our comments to the suggestions are in italics. In the manuscript the revisions are highlighted in yellow. More in detail:

 Comments and Suggestions for Authors by Reviewer #2

-       The authors should check the references listed in the text

-       The results of ecotoxicology tests could be presented graphically (suggestion)

-        some parts of the text should be corrected (see attachment)

Replay: Done! In particular the references have been reordered and the suggested corrections in the PDF file have been made. As regards ecotoxicology tests, the authors do not consider useful to present them graphically.